The prediction of swim start performance based on squat jump force-time characteristics

Thng Shiqi sthng@bond.edu.au 1 2
Pearson Simon 2
Rathbone Evelyne 1
Keogh Justin W.L. 1 3 4 5
1 Faculty of Health Sciences and Medicine, Bond University , Gold Coast , QLD , Australia
2 Queensland Academy of Sport , Nathan , QLD , Australia
3 Sports Performance Research Centre New Zealand, Auckland University of Technology , Auckland , New Zealand
4 Cluster for Health Improvement, Faculty of Science, Health, Education and Engineering, University of the Sunshine Coast , Sippy Downs , QLD , Australia
5 Kasturba Medical College, Mangalore, Manipal Academy of Higher Education , Manipal , Karnataka , India
Doyle Tim
Electronic publication date: 2020 Jun 1
Publication date: 2020
Volume: 8
Electronic Location ID: e9208
Received 2019 Nov 12; Accepted 2020 Apr 27
Copyright: ©2020 Thng et al.
Copyright year: 2020
Copyright holder: Thng et al.
License: This is an open access article distributed under the terms of the Creative Commons Attribution License, which permits unrestricted use, distribution, reproduction and adaptation in any medium and for any purpose provided that it is properly attributed. For attribution, the original author(s), title, publication source (PeerJ) and either DOI or URL of the article must be cited.
License URL: https://creativecommons.org/licenses/by/4.0/

Keywords: Swimming, Strength and conditioning, Resistance training, Swim start, Dry-land

Funding: Queensland Academy of Sport’s Sport Performance Innovation and Knowledge Excellence Unit in conjunction with Bond University Faculty of Health Sciences and Medicine This work was supported by the Queensland Academy of Sport’s Sport Performance Innovation and Knowledge Excellence Unit in conjunction with Bond University Faculty of Health Sciences and Medicine. The funders had no role in study design, data collection and analysis, decision to publish, or preparation of the manuscript.

==============================
Background

Depending on the stroke and distances of the events, swim starts have been estimated to account for 0.8% to 26.1% of the overall race time, with the latter representing the percentage in a 50 m sprint front crawl event (Cossor & Mason, 2001). However, it is still somewhat unclear what are the key physiological characteristics underpinning swim start performance. The primary aim of this study was to develop a multiple regression model to determine key lower body force-time predictors using the squat jump for swim start performance as assessed by time to 5 m and 15 m in national and international level swimmers. A secondary aim was to determine if any differences exist between males and females in jump performance predictors for swim start performance.

Methods

A total of 38 males (age 21 ±  3.1 years, height 1.83 ±  0.08 m, body mass 76.7 ±  10.2 kg) and 34 females (age 20.1 ±  3.2 years, height 1.73 ±  0.06 m, body mass 64.8 ±  8.4 kg) who had competed at either an elite (n = 31) or national level (n = 41) participated in this study. All tests were performed on the same day, with participants performing three bodyweight squat jumps on a force platform, followed by three swim starts using their main swimming stroke. Swim start performance was quantified via time to 5 m and 15 m using an instrumented starting block.

Results

Stepwise multiple linear regression with quadratic fitting identified concentric impulse and concentric impulse2 as statistically significant predictors for time to 5 m (R2 = 0.659) in males. With time to 15 m, concentric impulse, age and concentric impulse2 were statistically significant predictors for males (R2 = 0.807). A minimum concentric impulse of 200–230 N.s appears required for faster times to 5 m and 15 m, with any additional impulse production not being associated with a reduction in swim start times for most male swimmers. Concentric impulse, Reactive strength index modified and concentric mean power were identified as statistically significant predictors for female swimmers to time to 5 m (R2 = 0.689). Variables that were statistically significant predictors of time to 15 m in females were concentric impulse, body mass, concentric rate of power development and Reactive strength index modified (R2 = 0.841).

Discussion

The results of this study highlight the importance of lower body power and strength for swim start performance, although being able to produce greater than 200 or 230 N.s concentric impulse in squat jump did not necessarily increase swim start performance over 5 m and 15 m, respectively. Swimmers who can already generate greater levels of concentric impulse may benefit more from improving their rate of force development and/or technical aspects of the swim start performance. The sex-related differences in key force-time predictors suggest that male and female swimmers may require individualised strength and conditioning programs and regular monitoring of performance.

Introduction

Swim start performance has been identified as a determining factor for success, especially in sprint distance events, as it is the part of the race that the swimmer is travelling at the fastest velocity (Cossor & Mason, 2001; Tor, Pease & Ball, 2014). While the exact nature of starts may differ between the four swimming strokes, there are three primary phases that contribute towards the overall start performance. The block phase requires a quick reaction to the starting signal and a large take-off velocity that is primarily horizontal in direction (Garcia-Hermoso et al., 2013). The subsequent flight phase is an example of projectile motion, whereby the swimmer becomes airborne and finishes when they contact the water (Slawson et al., 2013; Tor, Pease & Ball, 2014). The flight phase is followed by the underwater phase, in which swimmers attempt to maintain a streamlined position with their arms outstretched in front of the head to minimise velocity loss while also performing multiple propulsive undulatory leg kicks (except in breaststroke) until their head resurfaces before the 15 m mark (Formicola & Rainoldi, 2015). The block, flight, and underwater phase account for approximately 11%, 5%, and 84% respectively of the total start time (Slawson et al., 2013). The average velocity during the underwater phase is highly dependent on the take-off velocity acquired in the block phase, the horizontal distance obtained in the flight phase, as well as the degree of streamlining and effectiveness of the undulatory leg kicks during the underwater phase (Tor, Pease & Ball, 2014).

As close margins often exist between medallists in sprint swimming events, being able to identify areas to achieve marginal gains in performance by tenths or even hundredths of a second can make a difference in overall performance (Bishop et al., 2009). Previous research has highlighted a key component of swim start performance is the ability to produce high forces off the starting block. In a recent systematic review of eight cross-sectional studies, Thng, Pearson & Keogh (2019) observed significant correlations between vertical jump and lower body strength scores to swim start performance in swimmers of a variety of standards, with these correlations typically higher for the jump than strength tests. Specifically, near perfect correlations (r > 0.90) between jump height or take-off velocity and swim start performance were observed in the eight studies. These results highlight the importance of lower body power and strength as an important component of swim start performance. However, out of the 8 cross-sectional studies identified in the systematic review (Thng, Pearson & Keogh, 2019), only one study utilised the OSB11 start block (OMEGA, Zurich, Switzerland) that is currently used in competitive swimming (Garcia-Ramos et al., 2016a). The OSB11 start block which was introduced by FINA in 2010 has an angled kick plate at the rear of the block that allows the swimmer to adopt a kick start technique. Honda et al. (2010) have identified that the angled kick plate on the OSB11 start block is capable of significantly improving both time to 5 m and 7.5 m, with a further 0.04 s improvement obtained in the kick start compared to the track start technique performed on the previous starting block. This is attributed to an increase in horizontal force application and subsequent take-off velocity from the additional contribution of the rear leg on the kick plate. This view of Honda et al. (2010) was consistent with the findings of Slawson et al. (2013) who observed higher peak horizontal and vertical force generation with the OSB11 start blocks in elite swimmers, with these forces significantly correlated to a better start performance as assessed by block time, take-off velocity and flight distance.

In addition, all of the studies described in the systematic review by Thng, Pearson & Keogh (2019) only involved correlational analyses. While correlations describe the nature of a relationship between two variables, other statistical approaches such as multiple linear regression may provide more information regarding what power and strength variables (hereafter referred to as force-time characteristics) of jumping performance that best predict swim start performance in high performance swimmers. The lack of research using the OSB11 start block and kick start technique in these correlation studies needs to be addressed, as this relative lack of ecological validity with the start technique used in seven of the eight published studies may limit the generalisability to contemporary high-performance swimming.

Another limitation of the previous literature is the small sample sizes of recreational to sub-elite swimmers (n = 7–27) and the relative focus on male swimmers at the expense of their female counterparts. This is a concern as previous research has established differences in force and power capabilities between males and females in other athletic activities (McMahon, Rej & Comfort, 2017; Rice et al., 2017). For example, a number of studies has observed that males are able to produce higher velocities at the same percent of one repetition maximum and have a greater rate of force development and countermovement jump height than females (Laffaye, Wagner & Tombleson, 2014; McMahon, Rej & Comfort, 2017; Rice et al., 2017; Torrejon et al., 2019). Rice et al. (2017) concluded that this greater jump height observed in males compared to females can be attributed to larger concentric impulse and thus greater velocity throughout most of the concentric phase at take-off in the countermovement jump. Further, the higher rate of force development and ability to produce greater velocities at the same percentage of one repetition maximum in males may be a result of greater muscle thickness and cross-sectional area, greater percentage of fast-twitch muscle fibres, greater concentration of anabolic hormones and higher neural activity during muscle contractions compared to females (Alegre et al., 2009). From a practical standpoint, these sex-related differences in force-time characteristics suggests there might need to be some potential differences in aspects of athletic monitoring and strength and conditioning programs between high-performance male and female swimmers.

The primary objective of this study was to develop a multiple regression model to determine key lower body force-time predictors for swim start performance using the squat jump in high performance swimmers. Considering the potential sex differences in force-time characteristics during jumping, a secondary aim was to determine if differences exists between males and females in jump performance predictors for swim start performance.

Materials & Methods

Study design

A cross-sectional study design was used to quantify the relationship between squat jump (SJ) force-time variables to swim start performance as assessed by times to 5 m and 15 m in national and international level swimmers. All tests were performed on the same day, with participants first performing SJ testing on the force platform, followed by a swim start performance test with a 30-minute recovery period in between each testing session.

Participants

Thirty-eight males and 34 females who had competed at either an elite (n = 31) or national level (n = 41) in front crawl, butterfly or breaststroke participated in this study. Backstroke was excluded due to the start being initiated from within the water, instead of on the elevated OSB11 starting block. Elite level swimmers comprised of swimmers who had competed internationally in either the Olympics, Commonwealth Games or World Championships. National level swimmers comprised of swimmers that have at least 2 years of experience in competing at a national level and competed at the most recent national championships. Swimmers were required to have at least 1 year of land-based resistance training experience under the supervision of a strength and conditioning coach. All participants gave written informed consent to participate in the study, which was conducted in accordance with the Declaration of Helsinki and approved by Bond University Human Research Ethics Committee (0000016006), The University of Queensland Human Research Ethics Committee (HMS17/41) and Swimming Australia Ltd.

Methodology

Squat jump test

Prior to the SJ test, participants completed a dynamic lower body warm-up under the supervision of a strength and conditioning coach. Following the warm-up, participants were given two practice jumps before the test was conducted. Jumps were performed on a force platform (ForceDecks FD4000, London, United Kingdom), with a sample rate of 1,000 Hz. Participants started in an upright standing position with their hands on their hips. They were then instructed to adopt a squat position using a self-selected depth that was held for 3 s before they attempted to jump as high as possible (Mitchell et al., 2017). A self-selected squat depth was chosen as it has been reported to produce the greatest jump height and higher peak force outputs in comparison to measured squat depths (Kirby et al., 2011). A successful trial was one that did not display any small amplitude countermovement at the start of the jump phase on the force trace (Sheppard & Doyle, 2008). All participants were asked to perform three maximal intensity SJ with a 30-second passive rest in between each effort.

The SJ trial with the highest jump height was kept for data analysis. Jump height was determined by the conventional impulse-momentum method (Jump Height = v2/2g, where v = velocity at take-off and g = gravitational acceleration) (Heishman et al., 2019). Ground reaction force data from the SJs were analysed using the commercially available ForceDecks software (ForceDecks, London, United Kingdom). Out of the variables provided by ForceDecks, 46 variables, excluding any left-to right asymmetry variables were initially extracted for use in further analysis. Descriptions of the SJ variables are provided in the Electronic Table S1.

Swim start

After completing a self-selected warm-up based on their usual pre-race warm-up routine, participants then performed three maximal effort swim starts with their main swim stroke (front crawl (n = 50), butterfly (n = 12), or breaststroke (n = 10)) while wearing their regular swim training swimsuits. Trials were started as per competition conditions and swimmers were instructed to swim to a distance past the 15 m mark, in order to ensure that representative values at the 15 m distance were obtained (Barlow et al., 2014). Two-minutes of passive recovery was given between each trial (Tor, Pease & Ball, 2015b). The start with the fastest time to 5 m for each individual with all swim strokes were selected for further analysis.

All 72 participants were included in the time to 5 m analysis irrespective of stroke performed, as the technical execution of the swim start does not differ until after 5 m. To avoid the potential confounding influence of the speed differences in both the underwater and swim phases of butterfly and breaststroke, only front crawl was included for time to 15 m analysis as it comprised of majority of the sample (n = 50).

Swim start performance were collected using a Kistler Performance Analysis System—Swimming (KPAS-S, Kistler Winterthur, Switzerland), which utilises a force instrumented starting block, constructed to match the dimensions of the Omega OSB11 block (KPAS-S Type 9691A1; Kistler Winterthur, Switzerland). Time to 5 m and 15 m were collected using five calibrated high speed digital cameras collecting at 100 frames per second, synchronised to the instrumented starting block using the KPAS-S. One camera was positioned 0.95 m above the water and 2.5 m perpendicular to the direction of travel to capture the start and entry of swimmer into the water, while the other three cameras were positioned 1.3 m underwater at 5 m, 10 m and 15 m perpendicular to the swimmer to capture the time to 15 m (Fig. 1) (Tor, Pease & Ball, 2015b). The times to 5 m and 15 m were defined as the time elapsed from the starting signal until the apex of the swimmers’ head passed the respective distances (Tor, Pease & Ball, 2015b). An Infinity Start System (Colorado Time Systems, Loveland, Colorado, USA) provided an audible starting signal to the athletes as well as an electronic start trigger to the KPAS-S system.

Figure 1 Overview of the camera set-up and instrumented starting block (Kistler Group, 2019).

Statistical analysis

Descriptive statistics are reported as mean ± SD for normally distributed continuous variables and frequency (%) for categorical variables. Normality was checked using histograms, normal Q-Q plots and the Shapiro–Wilk test. A principal component analysis (PCA) was used to identify optimal sets of key performance indicators on the 46 jump variables extracted from ForceDecks force platform (ForceDecks, London, United Kingdom). This method has been used in previous studies that sought to identify kinematic and kinetic predictors of athletic performance from a number of highly interrelated vertical jump performance measures (Kollias et al., 2001; Laffaye, Bardy & Durey, 2007). A second PCA was conducted to explore the reduced dataset of 32 jump performance variables and identify the principal components (PC) summarising the primary force-time variables. The decision on a suitable number of PCs to retain in each PCA required eigenvalues of 1.0 or greater (Kaiser criterion) and was supported using a scree plot.

Multiple linear regression models using a stepwise regression method were initially performed to identify the potential predictors of the outcome variables of time (s) to 5 m and 15 m. Analyses were carried out on the entire dataset, and also on the data split by sex. Second order polynomial models were also investigated, as visual inspection identified that these quadratic models better matched the data for males than the linear models, with this also confirmed by significantly higher R2 for the quadratic models (Bobrovitz & Ottenbacher, 1998; Park, Marascuilo & Gaylord-Ross, 1990). Collinearity diagnostics were used to avoid the problem of multicollinearity. The assumptions of normality, linearity and homoscedasticity of residuals were verified. Results of the regression modelling are presented in terms of unstandardized coefficients, the 95% CI and p-values, along with the R2 and standard error of estimate. Data were analysed with statistical software R version 3.5.3 and SPSS version 23.0.0 (SPSS Inc., Chicago, IL). P-values less than 0.05 were deemed to indicate statistical significance.

Results

Seventy-two swimmers, comprising 38 males and 34 females were included in this study. The physical characteristics of the participants are described in Table 1. Out of the 72 participants, 50 participants performed the swim start using the front crawl technique, with an additional 12 participants performing butterfly and 10 participants using breaststroke. Statistically significant differences among males and females were observed in a number of variables (Table 1), with males significantly heavier, taller and faster to 5 m and 15 m than females.

Table 1 Physical characteristics of participants.

All data is presented as means and standard deviations.

Variables	Males	Females	
	5 m (n= 38)	15 m (n= 26)	5 m (n= 34)	15 m (n= 24)	
Age (years)	21.0  ± 3.1	21.2  ± 3.2*	20.1  ± 3.2	19.2  ± 3.2	
Body mass (kg)	76.7  ± 10.2**	76.5  ± 11.0**	64.8  ± 8.4	64.2  ± 8.4	
Height (m)	1.83  ± 0.08**	1.85  ± 0.08**	1.73  ± 0.06	1.73  ± 0.06	
Time to 5 m (s)	1.48  ± 0.09**		1.65  ± 0.08		
Time to 15 m (s)		6.4  ± 0.44**		7.3  ± 0.5	
Notes.

* p < 0.05.

** p < 0.001 between males and females.

In the first PCA analysis on the 46 jump variables extracted from ForceDecks force platform (ForceDecks, London, United Kingdom), four PCs which explained 82% of the variance were identified. Thirty-two most influential jump variables were identified from this initial PCA. A secondary PCA was run to explore the new dataset of 32 jump performance variables. The first three components, which explained 93% of the variance, were retained. From this set, 15 variables were identified as potential predictors in subsequent regression models (Table 2). The results revealed that Component 1 accounting for 67.5% of the variance, was of predominantly kinetic component. Component 2 accounting for 17.1% of the variation, was predominantly a time-dependent kinematic component. Lastly, Component 3 accounted for 8.5% of the variation, with the highest load attributed to bodyweight.

Table 2 List of 15 most influential potential predictors of swim start performance identified from the PCA and their correlations with the principal components.

Potential predictors	Principal Component	
	PC1	PC2	PC3	
Variation explained for each component	67.5%	17.1%	8.5%	
Bodyweight (BW)	−0.71	0.11	0.68	
Concentric impulse	−0.88	0.31	0.34	
Concentric mean force	−0.91	−0.09	0.39	
Concentric mean power	−0.94	0.13	0.14	
Concentric peak force	−0.92	−0.15	0.32	
Concentric rate of power development (RPD)	−0.93	−0.31	0.04	
Force at peak power	−0.92	−0.05	0.33	
Peak power	−0.95	0.24	0.14	
Reactive strength index modified (RSImod)	−0.90	−0.12	−0.20	
Take-off peak force	−0.92	−0.15	0.32	
Concentric peak velocity	−0.77	0.55	−0.29	
Concentric rate of force development (RFD) BW	−0.59	−0.75	−0.15	
Concentric RFD	−0.72	−0.66	0.05	
Jump height (impulse-momentum)	−0.75	0.56	−0.31	
Velocity at peak power	−0.68	0.66	−0.27	

Linear stepwise multiple regression analyses were performed using the ForceDecks SJ data to predict time to 5 m (see Fig. 2 and Table 3) and time to 15 m (see Fig. 3 and Table 4) in the overall sample of males and females as well as male and female subgroups.

Figure 2 Relationship between concentric impulse (N.s) against time to 5 m (s) across females and males. Poly means polynomial regression to order 2, i.e., quadratic.

The grey dotted line and diamond markers represent the linear relationship between concentric impulse and time to 5 m in females. The dashed line with circle markers represents the quadratic relationship between concentric impulse and time to 5 m in males.

Table 3 Multiple linear regression models to predict swim start time (s) to 5 m performance in females, males and both females and males combined.

		% contribution	Beta coefficient (95% CI)	p- value	
All	Concentric Impulse (N.s)	70.4	−0.002 (−0.002 to −0.001)	<0.001	
	Sex (Females)	5.4	0.065 (0.028 to 0.102)	0.001	
	RSImod (m/s)	1.5	−0.084 (−0.164 to −0.004)	0.040	
	Constant		1.882 (1.790 to 1.974)	<0.001	
	R2 (SEE)		0.773 (0.059)		
Females	Concentric impulse (N.s)	51.6	−0.003 (−0.004 to −0.002)	<0.001	
	RSImod (m/s)	9.5	−0.209 (−0.315 to −0.104)	<0.001	
	Concentric Mean Power (W)	7.8	0.0002 (0.00004 to 0.0003)	0.010	
	Constant		2.103 (1.986 to 2.219)	<0.001	
	R2 (SEE)		0.689 (0.047)		
Males	Concentric Impulse (N.s)	53.6	−0.010 (−0.015 to −0.005)	<0.001	
	Concentric Impulse2 (N.s)2	12.3	0.00002 (0.00001 to 0.00003)	0.001	
	Constant		2.645 (2.167 to 3.124)	<0.001	
	R2 (SEE)		0.659 (0.055)		
Notes.

SEE, standard error of estimate.

Figure 3 Relationship between concentric impulse (N.s) against time to 15 m (s) across females and males. Poly means polynomial regression to order 2, i.e., quadratic.

The grey dotted line and diamond markers represent the linear relationship between concentric impulse and time to 15 m in females. The dashed line with circle markers represents the quadratic relationship between concentric impulse and time to 15 m in males.

Table 4 Multiple linear regression models to predict swim start time (s) to 15 m performance in females, males and both females and males combined.

		% contribution	Beta coefficient (95% CI)	p value	
All	Concentric Impulse (N.s)	76.1	−0.008 (−0.011 to −0.004)	<0.001	
	Age (years)	3.5	−0.052 (−0.087 to −0.018)	0.004	
	Sex (female)	3.0	0.362 (0.151 to 0.572)	0.001	
	Constant		9.074 (8.503 to 9.646)	<0.001	
	R2 (SEE)		0.826 (0.278)		
Females	Concentric Impulse (N.s)	65.1	−0.030 (−0.041 to −0.020)	<0.001	
	Body mass (kg)	9.3	0.035 (0.006 to 0.064)	0.020	
	Concentric RPD (W/s)	4.9	0.0002 (0. 00006 to 0.0003)	0.004	
	RSImod (m/s)	4.8	−1.714 (−3.215 to −0.213)	0.027	
	Constant		9.303 (8.398 to 10.208)	<0.001	
	R2 (SEE)		0.841 (0.225)		
Males	Concentric Impulse (N.s)	66.6	−0.033 (−0.058 to −0.008)	0.011	
	Age (years)	9.4	−0.048 (−0.086 to −0.010)	0.016	
	Concentric Impulse 2 (N.s)2	4.7	0.00007 (0.000007 to 0.0001)	0.031	
	Constant		11.188 (8.975 to 13.401)	<0.001	
	R2 (SEE)		0.807 (0.205)		

Time to 5 m

The scatterplot in Fig. 2 shows a quadratic relationship between SJ concentric impulse and time to 5 m in males (R2 = 0.693). For a fast time to 5 m for males, visual inspection of the data suggests a minimum concentric impulse production of around 180–200 N.s is required. While visual inspection of the model suggested no additional reduction in time to 5 m with a higher concentric impulse for most swimmers, there are some outlier individuals who appear to derive additional performance benefit from an increased concentric impulse up to approximately 230 N.s. The relationship between concentric impulse and time to 5 m observed in females was linear (R2 = 0.487), but this relationship was affected by other factors outlined in Table 3.

Concentric impulse was a statistically significant predictor in all three regression models (Table 3). The best prediction equations for time to 5 m in females and males were as follows: Females:T5 m (s)=2.103−0.003concentric impulse−0.209RSImod+0.0002concentric mean power

Males:T5 m (s)=2.645−0.010concentric impulse+0.00002concentric impulse2.

Time to 15 m

The scatterplot in Fig. 3 shows a quadratic relationship between SJ concentric impulse and time to 15 m in males (R2 = 0.746). For a fast time to 15 m in males, a minimum concentric impulse production of around 230 N.s is required. However, similar to Fig. 2, the relationship between concentric impulse and time to 15 m observed in females was linear (R2 = 0.651) but this relationship was also affected by other factors presented in Table 4.

The SJ concentric impulse was also the main significant predictor in all three regression models of the time to 15 m (Table 4). The best regression models were as follows: Females:T15 m (s)=9.303−0.030concentric impulse+0.035bodymass+0.0002concentric RPD−1.714RSImod

Males:T15 m (s)=11.188−0.033concentric impulse−0.048age+0.00007concentric impulse2.

Discussion

The present study revealed that several lower body force-time characteristics, in particular concentric impulse, were significantly related to swim start performance in national and international level swimmers. However, when these analyses were performed for each sex individually, several differences in the prediction of swim start performance were observed. These sex-related differences in key force-time characteristics suggest that strength and conditioning programs and regular monitoring of performance may need to be tailored to male and female swimmers.

In the swim start, swimmers have to apply large forces rapidly on the start block to maximise horizontal take-off velocity, which in turn allows them to travel farther horizontally in the air before entering the water (Rebutini et al., 2014). This task demand is consistent with the impulse-momentum relationship, whereby an impulse (the product of force and time of force application) needs to be generated to cause a change in momentum (i.e., velocity) of the system (Schilling, Falvo & Chiu, 2008). An analysis by Tor, Pease & Ball (2015a) of the above water parameters in the swim start have found that take-off velocity and time on block were key predictors of swim start performance as assessed by time to 15 m using the OSB11 start block. Strong positive correlations between peak forces in the countermovement jump and peak forces on the OSB11 start block have also been reported by Cossor and colleagues (Cossor et al., 2011). Thus, to be able to achieve a high take-off velocity, a swimmer needs to be able to apply high forces/ impulses off the starting block. Given that the swim start is mainly a concentric only movement, the findings of the present study further emphasise the important association between a swimmers’ ability to produce impulse in the SJ and swim start performance.

It was expected that the current study would demonstrate a stronger prediction to 5 m than 15 m in the swim start. This hypothesis was based on how the movement pattern in the SJ is similar to the initial push-off in the block phase as well as the findings of Garcia-Ramos et al. (2016b) and Benjanuvatra, Edmunds & Blanksby (2007), who reported a significant correlation in take-off velocity (Garcia-Ramos et al., 2016b) and jump height (Benjanuvatra, Edmunds & Blanksby, 2007) in the SJ to 5 m (r =  − 0.56 and r =  − 0.92 respectively) but not 15 m. In contrast to this initial hypothesis, the current study demonstrated that the SJ force-time variables explained a greater amount of variance in time to 15 m than time to 5 m. Results of the current study were also consistent with Garcia-Ramos et al. (2016a) who observed that the correlations between jump height and swim start performance were greater for the time to 15 m (r =  − 0.67) than time to 5 m (r =  − 0.55) using the kick start technique. Such equivalence in the literature was surprising, but it is possible that these contrasting findings from the current study to the limited literature could be attributed to a variety of between study differences, including the swim start technique and start block, as well as the sample size and homogeneity of participants included in the previously published studies. The current study utilised the kick start technique on the OSB11 start block, whereas Benjanuvatra, Edmunds & Blanksby (2007) and Garcia-Ramos et al. (2016b) utilised the grab start and track start, respectively. In addition, both of these studies included only female swimmers and had substantially smaller sample sizes (n = 20 and n = 7), whereas the current study utilised a mix of male and female swimmers, with a larger sample size for both time to 5 m (n = 72) and 15 m (n = 50). As previously mentioned, the underwater phase is a key parameter in swim start performance, as a swimmer spends the highest percentage of the start in the underwater phase for all swim strokes (Cossor & Mason, 2001; Tor, Pease & Ball, 2015a; Vantorre et al., 2010). Garcia-Ramos et al. (2016b) have suggested that swimmers require high levels of lower body strength and power to maximise their underwater kick performance. Therefore, it is possible that the stronger prediction in time to 15 m than 5 m in this study and the study by Garcia-Ramos et al. (2016a) may reflect the commonality in lower body force-time characteristics required for the block phase with the kick start technique and the undulatory kicks performed during the underwater phase.

Another focus of this study was examining potential sex-related differences in the force-time characteristics that may underpin swim start performance in high-performance swimmers. While concentric impulse was the strongest predictor for time to 5 m and 15 m in both males and females, the current study identified some differences between the sexes with respect to the predictors of time to 5 m and 15 m. For a quick time to 5 m and 15 m in males, a minimum concentric impulse of 200–230 N.s appears required, with any additional impulse production not being associated with a reduction in swim start times for most male swimmers. However, it is worth noting that within the dataset, there appear to be some athletes whose performance sits outside the generalised trend, showing increased performance gains from additional concentric impulse about the level at which most individuals are deriving no further benefit (Figs. 2 and 3). Nevertheless, these findings tend to suggest that for male swimmers capable of producing greater than 230 N.s of impulse, it might be most beneficial for their strength and conditioning program to focus on improving their rate of force development, as it is possible that developing this high level of impulse in a shorter block time is required to further improve their swim start performance.

In contrast to the results for the male swimmers, which had concentric impulse as the sole contributing force-time variable from squat jumps, the swim start performance to 5 m and 15 m for females were also influenced by other factors such as RSImod, mean power and concentric RPD. A few possible explanations for the differing strategies could be attributed to maximal strength capacity, load-velocity and neuromuscular capability between both sexes. Although lower body muscular strength was not measured in the current study, maximal strength has been shown to be a limiting factor in jumping ability and other lower body measure of explosive strength (Andersen & Aagaard, 2006; Suchomel et al., 2018). Previous research has demonstrated that males possess greater maximal strength and ability to produce greater velocities at the same percentage of one repetition maximum than their female counterparts (Sole et al., 2018; Torrejon et al., 2019). When comparing the force-time curves in the countermovement jump between sexes, previous research has reported that the male and female differences in countermovement jump height were attributed to force characteristics and not temporal characteristics of the force-time curve (Beckham et al., 2019; Sole et al., 2018). This suggests that both sexes possess similar abilities to express forces, but the primary difference in jumping ability was due to the rate and magnitude of force production during both peak eccentric and concentric force production, which may be explained by differences in muscle architecture and structure, such as thickness and size of muscle fibers (Laffaye, Wagner & Tombleson, 2014). These sex related differences might therefore explain some of the differing swim start predictors identified in the present study.

Previous studies have suggested that there is a trade-off between time spent on the starting block and take-off velocity, as the likelihood of greater impulses being produced with greater block times (Breed & McElroy, 2000; Takeda et al., 2017; Vantorre, Chollet & Seifert, 2014). From a practical standpoint, a possible strategy to increase impulse generated on the starting block without excessively increasing the time of force application is to increase muscular strength and rate of force development qualities of the lower body through heavy resistance training, ballistic concentric-dominant exercises (i.e., jumps without a preceding eccentric contraction) and plyometric training (Aagaard et al., 2002; West et al., 2011). Heavy resistance training has been shown to increase power production, rate of power development, rate of force development and increases in muscle fiber cross-sectional area and neuromuscular activity (Jakobsen et al., 2012). Ballistic/ plyometric training may improve the transfer of maximal strength to power production and rate of force development (Suchomel et al., 2018), thereby significantly improving swim start performance metrics including time to 5 m, take-off velocity and impulse (Bishop et al., 2009; Rebutini et al., 2014; Rejman et al., 2017). From a monitoring perspective, if a swimmer possesses the concentric impulse production required but has slow start times to 5 m and 15 m, improving rate force development and/or assessing technical factors such as angle of entry, degree of streamline, hydrodynamic drag and underwater propulsion may be imperative to maximise strength transfer to the swim start and ultimately swimming performance (Vantorre, Chollet & Seifert, 2014). Thus, swimmers should be concurrently performing lower body strength and conditioning program that includes some mixture of strength, ballistic and/or power training while ensuring sufficient practice of the swim start to optimise the transfer of their strength and conditioning program in improving swim start performance (Breed & Young, 2003).

There are some limitations in this study that could be addressed in future research. Firstly, baseline strength was not measured in any of the participants. Future work should examine the relationship between lower body force-time characteristics in strength matched swimmers and its effect on swim start performance to elucidate if differences between male and female swimmers were due to muscular strength or neuromuscular differences (Nimphius, 2019). Secondly, due to the difference in sample sizes for the different swim strokes in the current study, it would be worth exploring what force-time characteristics underpin swim start performance in other swim strokes in comparison to the front crawl, and if there are different neuromuscular qualities required for swim start performance in the different swim strokes.

Conclusion

In summary, this study has identified bodyweight squat jump concentric impulse as a key lower body force-time characteristic that was significantly related to swim start performance in high-performance swimmers. As impulse is the product of the ground reaction force and time of force application, it is integral for a swimmer to have the requisite ability to generate a high level of concentric impulse in a relatively short amount of time. Due to the different strength of the prediction equations, it appears that male and female swimmers utilise somewhat differing strategies during the swim start. While it is unknown if this is predominantly a result of the differences in muscular strength and force producing capacity between sexes, our results highlight the need for strength and conditioning coaches to consider individualising training programs to enhance swim start performance and ultimately swimming performance between sexes.

Supplemental Information

Table S1 Definition of squat jump variables obtained from the ForceDecks force platform

Click here for additional data file.

Supplemental Information 1 Raw data file for squat jumps and time to 5 m and 15 m used in data analysis

Click here for additional data file.

The authors would like to acknowledge Dr. Mark Osborne for his support in preparing this manuscript.

Additional Information and Declarations

Competing Interests

Author Contributions

Human Ethics

Data Availability

Justin Keogh is an Academic Editor for PeerJ.

Shiqi Thng conceived and designed the experiments, performed the experiments, analyzed the data, prepared figures and/or tables, authored or reviewed drafts of the paper, and approved the final draft.

Simon Pearson conceived and designed the experiments, performed the experiments, analyzed the data, authored or reviewed drafts of the paper, and approved the final draft.

Evelyne Rathbone analyzed the data, authored or reviewed drafts of the paper, and approved the final draft.

Justin W.L. Keogh conceived and designed the experiments, analyzed the data, authored or reviewed drafts of the paper, and approved the final draft.

The following information was supplied relating to ethical approvals (i.e., approving body and any reference numbers):

Ethical approval to conduct this study was attained from Bond University Human Research Ethics Committee (0000016006), The University of Queensland Human Research Ethics Committee (HMS17/41) and Swimming Australia Ltd.

The following information was supplied regarding data availability:

The raw data is available in the Supplementary Files.

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
