# Peer review of "The prediction of swim start performance based on squat jump force-time characteristics"

_PeerJ, doi:10.7717/peerj.9208_

## Round 0.1 · original submission · Minor Revisions

Thank you for your submission to PeerJ. Please address the reviewers' comments in detail prior to resubmission. I look forward to your next version.

·

Basic reporting

The article in general is written well and is technically correct. I would suggest that you ensure that the references are in the format recommended by the journal. The year appears to be in the wrong position and places should have capital letters (e.g. Sydney). The mix of swimming starts, jumping techniques and gender appear to be appropriate in the reference section.

https://peerj.com/about/author-instructions/

The figures are relevant and appropriately labelled with all data supplied for review.

Experimental design

With the equipment that was used within the study I was looking for force information from the starting block, but it appears that the only comparisons made were between the squat jumps and times to 5m or 15m.

The logic behind the study was sound but it does feel like similar work has been conducted previously by other authors. Is there some way that you could add the block data to the regression analysis?

Information on each of the key variables would be useful for clarification purposes. Further notes are presented in the general comments section.

Validity of the findings

This research provides conclusions based on starts with larger subject numbers than previous research. I would question why the 15m value was not used for all strokes in order to add to the current research. Male/female differences have been reported previously but it is good that this research has validated the finding with a large group of elite swimmers.

Additional comments

Line 5 – close margins between success and what?
Line 9 – primary is used in two consecutive sentences
Line 15 – insert the word “and” before 34
Line 22 – Look to rephrase the sentence to something like “Faster times to 5 m and 15 m in males”
Line 27 – consider changing the start of the sentence to “Variables impacting time to 15 m”
Line 34 – you mention rate of force development – was this investigated? There are some results in your raw data table but I could not find any mention of it in the discussion.
Line 36 – check that you are using US spelling as required for this journal (individualized)
Line 48 – breaststrokers are legally allowed to perform one undulatory fly kick so you may wish to consider referring to multiple kicks in the other strokes
Line 58 – should you have the reference after Thing et al rather than at the end of the sentence?
Line 65 – US spelling for utilized
Line 67 – use correct reference (FINA)
Line 73 – was there any impact from changes to the angle of the block between the old and new design?
Line 82 – change “may” to “that”
Line 84 – what do you mean by ecological validity
Line 87 – are these subject numbers correct? I have not been able to access the article that you are referring to so am unclear of the 8 papers that were reviewed
Line 101 – were these sex-related differences surprising? Could you be stronger in your wording? What differences have been used to assess muscle fibre type within genders?
Line 118 – this sentence is repeated from the introduction/abstract
Line 124 – how do you define regularly?
Line 132 – where do you identify the variables used in the analysis? What is concentric impulse 2?
Line 150 – look to rephrase this sentence to something like “46 variables provided by ForceDecks were initially extracted for use in the analysis process”
Line 159 – why did you use different protocols for the different strokes? Previous research has identified similarities in the underwater phase for backstroke, freestyle and butterfly so it may have been possible to group the information or make note to the differences in the strokes for the sample that you used.
Line 194 – look to rephrase the sentence commencing with second order polynomial
Line 208 – make reference to other papers that have found that males were significantly heavier, taller and faster than females
Line 234 – have you described how concentric impulse is calculated? You should also be clear that this is during the squat jump rather than on the block.
Line 241 – for clarification the sample size is smaller for time to 15 m so why include table 3 in this section of the results?
Line 257 – can you be more specific than “around 230 N.s”?
Line 259 – change the word “outlined” to “presented”
Line 276 – why only “may”?
Line 278 – US spelling for maximize
Line 292 – if one of the aims of the study was to determine the level of prediction between the 5 m and 15 m start times, why did you only measure 15 m times in freestyle?
Line 310 – is the comment related to underwater phase valid for all strokes and distances?
Line 352 – time of force application is important but there is no reference to this time on the block
Line 360 – does this sentence have any relevance to concentric training?
Line 367 – US spelling for utilize

Reviewer 2 ·

Basic reporting

This article is generally well written. The language is clear and consice and the findings are clear to understand. I have made a few suggestions for improvement below.

Experimental design

It was not clear that the other strokes were excluded from 15 m analysis.

Was the different stroke data reported in te 5 m analysis? It was not in the discussion.

Why was 7.5 and 10 m time not considered as well?

Did you consider only using dive and glide trials?

Does the self-selected swuat jump depth affect overall jump height? please provide evidence to support this.

No limitations to the current study design were discussed?

Is there a limit on the number of tables and figures for this journal?

Provide some supporting evidence around the statistical analysis used.

Why did you not use a weight squat jump?

Was the time between jumps testing and swim start testing always consistent?

Were the trials suited ir unsuited?

Validity of the findings

The data analysis is strong and the findings of this study are novel. But I would have liked to see more data on the baseline strength of each of the athletes as I think the main differenes in Sex exist because females are in general weaker than males. Please dicuss this more...

Also discuss...if females are weaker could a tactic be to increase their strength first and then their start predictors would be similar to males?

Additional comments

First few lines in the abstract...this is fairly obvious. Please remove. Could maybe add something like "The start has been shown to contribute to XXX about of total race time" instead.

If word count permits, could add a bit more detail around the methodology in the methods secton of the abstract.

Line 53 - replace ref number 6 with a more relevant referene that utilises the kick start.

Line 86 - consider a new paragraph

Think about splitting the conclusion into major findings and practical applications.

Try to insert references at the end of the sentence to enhace flow of the manuscript.

Overall, I well written and constructed study. Great Work!

---

## Round 0.2 · Minor Revisions

Thank you for your revision of this paper. As you can see from the reviewer comments the paper is now much improved from the original submission. One reviewer has identified a few minor items which need to be addressed before publication. Please take these into consideration when revising the paper. Thank you again.

·

Basic reporting

The information presented in the revised version of the article meets the PeerJ standard of reporting.

Changes were made to the referencing but there is one reference that uses capital letters for the title.

Experimental design

The research questions are well defined with the gap identified from the author's earlier article reviewing current literature.

Validity of the findings

No comment

Additional comments

The work that the author has done in reviewing the original paper is to be commended.

Reviewer 2 ·

Basic reporting

Thank you for addressing all of my previous comments. It is clear that the authors have thoroughly considered all feedback and made the changes needed.

The article is well written and will contribute to body of knowledge on swimming start performance.

Experimental design

The added detail in the methods sections has made the experimental design much clearer.

My main concern is that there is some research out there (Tor. 2014) that has explored the contribution of the free swimming phase to the Start performance. Given that all swimmers must resurface before 15 m there is some element of free swimming involved when assessing time to 15 m. The contribution of free swimming obviously changes with race distance and individual under water kick ability etc. I am wondering if the results would change if you looked at time to 10 m or time to 7.5 m in addition or instead?

Please address this either in your methods or discussion phase as I think this is quite important given there is a lot of things that can occur to affect over time performance.

Validity of the findings

The practical applications are not much clearer and the research questions/aims the authors have started are addressed well within the article.

Additional comments

Here are some typos that I picked up.
Table 2 - think about adding units.
Line 96 - change have to has
Line 110 - Remove therefore
Line 231 - check PeerJ formatting guidelines but I am unsure there is allowed to be a space between number and percent sign.
Line 253 - What is meant by "visual inspection"? Is this reliable and valid? Please provide more details.
Sentence starting on Line 370 - please reword its very repetitive.
Line 384 - The sentence starting on this line is very long. Please consider rewording.

---

## Round 0.3 · accepted · Accept

Thank you for taking the time to make considered changes to the manuscript based on reviewer feedback. I think you will agree that this process has helped to improve the paper.